# Weighted Gene Co-Expression Network Analysis Identifies a Functional Guild and Metabolite Cluster Mediating the Relationship between Mucosal Inflammation and Adherence to the Mediterranean Diet in Ulcerative Colitis

**DOI:** 10.3390/ijms24087323

**Published:** 2023-04-15

**Authors:** Jaclyn C. Strauss, Natasha Haskey, Hena R. Ramay, Tarini Shankar Ghosh, Lorian M. Taylor, Munazza Yousuf, Christina Ohland, Kathy D. McCoy, Richard J. M. Ingram, Subrata Ghosh, Remo Panaccione, Maitreyi Raman

**Affiliations:** 1Department of Medicine, Cumming School of Medicine, University of Calgary, 2500 University Drive NW, Calgary, AB T2N 1N4, Canada; 2Department of Biology, Irving K Barber Faculty of Science, University of British Columbia—Okanagan, 3137 University Way, Kelowna, BC V1V 1V7, Canada; 3International Microbiome Centre, HRIC 4AA08 Foothills Campus, University of Calgary, Calgary, AB T2N 1N4, Canada; 4APC Microbiome Ireland, College of Medicine and Health, University College Cork, College Road, National University of Ireland, T12 K8AF Cork, Ireland

**Keywords:** inflammatory bowel disease, ulcerative colitis, diet, microbiome, metabolomics, weighted gene co-expression analysis (WGCNA)

## Abstract

Diet influences the pathogenesis and clinical course of inflammatory bowel disease (IBD). The Mediterranean diet (MD) is linked to reductions in inflammatory biomarkers and alterations in microbial taxa and metabolites associated with health. We aimed to identify features of the gut microbiome that mediate the relationship between the MD and fecal calprotectin (FCP) in ulcerative colitis (UC). Weighted gene co-expression network analysis (WGCNA) was used to identify modules of co-abundant microbial taxa and metabolites correlated with the MD and FCP. The features considered were gut microbial taxa, serum metabolites, dietary components, short-chain fatty acid and bile acid profiles in participants that experienced an increase (*n* = 13) or decrease in FCP (*n* = 16) over eight weeks. WGCNA revealed ten modules containing sixteen key features that acted as key mediators between the MD and FCP. Three taxa (*Faecalibacterium prausnitzii, Dorea longicatena, Roseburia inulinivorans*) and a cluster of four metabolites (benzyl alcohol, 3-hydroxyphenylacetate, 3-4-hydroxyphenylacetate and phenylacetate) demonstrated a strong mediating effect (ACME: −1.23, *p* = 0.004). This study identified a novel association between diet, inflammation and the gut microbiome, providing new insights into the underlying mechanisms of how a MD may influence IBD. See clinicaltrials.gov (NCT04474561).

## 1. Introduction

Inflammatory bowel disease (IBD), Crohn’s disease (CD) and ulcerative colitis (UC) are a group of chronic, relapsing and remitting conditions affecting the gastrointestinal tract [1]. The pathogenesis of IBD remains unclear; however, evidence suggests that it results from a complex interaction between genetic risk factors, an aberrant host immune response, alterations in the gut microbiome, and environmental factors, including diet [1]. Despite advances in pharmaceutical approaches that have contributed to improved outcomes, including steroid-free clinical and endoscopic remission, current treatments for UC may not lead to lasting remission [2]. Yet diet is of interest to patients, and emerging evidence has demonstrated that dietary interventions are associated with disease activity and inflammation in IBD through their effects on the gut microbiome [3]. The causal mechanisms of how diet influences the gut microbiome and metabolome are poorly understood. Thus, there is an urgent need to understand the complex interaction between diet, the gut microbiome, metabolome and the host immune system to apply diet as a potential therapeutic tool for managing IBD.

The rise in -omics technologies has improved our understanding of IBD, as IBD is known to be a complex disease composed of many interactions between the host and the gut microbes [4]. Dysbiosis, defined as an alteration of intestinal microbes relative to abundance and diversity, is a hallmark of IBD and leads to alterations in key metabolic pathways, as well as alterations in the gut metabolome [5,6]. A reduced abundance of health-associated microbes, including the butyrate-producing bacteria *Faecalibacterium prausnitzii* and *Roseburia* spp., along with an increased abundance of pathobionts from the phylum Proteobacteria has been reported in IBD [5,6]. The IBD metabolome is characterized by reduced levels of short-chain fatty acids (SCFAs) such as butyrate, altered levels of amino acids, notably the branched-chain amino acids, fatty acid esters [acylcarnitines] and changes in secondary bile acid profiles [7]. Despite the rapid increase in research in the -omics field, we are only beginning to unravel the complexities of how specific dietary patterns may influence microbial growth and metabolite production in the context of IBD [8,9].

Results from clinical studies on the Mediterranean diet (MD) provide evidence of the interconnection between diet, microbes and their metabolites and its role in preventing and managing several chronic inflammatory diseases [10,11,12,13,14], including IBD [15,16]. Despite correlational evidence that dietary patterns are an important environmental factor that can alter the microbiome and metabolome composition and functionality [17], causal evidence demonstrating an association between diet, microbial species, metabolites, and mucosal inflammation is limited [8,9].

In health research, correlation networks are increasingly used to integrate multi-omics data (e.g., shotgun metagenomics, metabolomics, etc.) with host clinical traits (e.g., diet, age, BMI, medications, biomarkers, etc.) [18]. While originally created for computing gene co-expression networks, weighted gene co-expression network analysis (WGCNA) is now being used in studies to integrate multi-omics data [19]. The primary objective of WGCNA is to identify strongly correlated modules across different data types and then associate these modules with clinical traits, which may enable the discovery of potential biomarkers that we can target through dietary interventions [18].

In this study, we applied a network-based approach using WGCNA to identify host–microbe interactions between microbial taxa, dietary components, serum and stool metabolites, and SCFA and bile acid profiles associated with the MD and changes in fecal calprotectin (FCP) in adults with UC. We identified mediation networks that indicate a stepwise mechanistic trajectory between the MD and FCP that is potentially mediated by specific microbiome taxa and metabolites.

## 2. Results

### 2.1. Participant Characteristics Recruited to the Open-Label Clinical Study

The clinical study results are reported in Appendix A. Briefly, 40 participants (*n* = 22 diet intervention and *n* = 18 habitual diet) completed the 8-week study. The median age of the participants was 37 years (range 21–80 years), with a median BMI of 26 Kg/m^2^ (range 19–32 Kg/m^2^) (Appendix A). Most participants had active disease at baseline based on PMS, median PMS = 3 (IQR 1–5); fifteen (35%) remission (PMS 0–1), twelve (31%) mild (PMS 2–4), seven (18%) moderate (PMS 5–6) and five (13%) severe disease (PMS 7–9). There was a statistically significant decrease in sulfur intake in the MD intervention from baseline to week 8 (*p* = 0.003), but it was not different between groups nor within the habitual diet (Appendix A). No changes in MDS or FCP were observed within or between groups. Valerate (SCFA) and glycochenodeoxycholic acid (bile acid) were significantly different between groups at baseline and week 8 (*p* = 0.05 and *p* = 0.02, respectively). No significant differences were observed between the rest of the SCFAs (Appendix A) or bile acids (Appendix A) within or between groups (*p* < 0.05). Marginal improvements in PMS (median 2.0) were observed from baseline and week 8 in participants following the intervention diet (*p* = 0.003); however, this also occurred in the habitual diet (*p* = 0.007). In summary, reducing dietary sulfur intake with a Mediterranean-like diet pattern did not detect clinically significant differences (defined as mean difference in PMS of three points) over the habitual diet (control). As we noted some interesting trends in the data, we proceeded to conduct a post hoc analysis of our study population.

We identified two distinct trajectories in FCP, where participants either increased or decreased FCP across time points (Figure 1); therefore, we completed a post hoc analysis using WGCNA. We included participants from whom we had dietary assessment data, metabolomic data, bile acid profiles, SCFA profiles and participants that had either an increase or decrease in FCP across time points with matching gut microbiome profiles (*n* = 29) to examine which features might predict inflammation (FCP).

### 2.2. Characteristics of Participants with a Change in FCP across Time Points

Twenty-nine participants had a change in FCP across time points (baseline to week 8). The baseline characteristics of the two cohorts whose FCP increased (*n* = 13) versus decreased (*n* = 16) from baseline to week 8 are shown in Table 1. When comparing disease activity (PMS) at baseline, in the cohort whose FCP increased over the eight weeks, six of thirteen participants (46%) were in clinical remission, and one of thirteen (8%) had clinically severe disease. In the cohort whose FCP decreased from baseline over the eight weeks, four of sixteen (25%) were in clinical remission, and two of sixteen (13%) had clinically severe disease. There was a statistically significant change in median FCP level from baseline (56 mcg/g [IQR 0–551]) to week 8 (961 mcg/g [IQR 217–2436]) (*p* = 0.001) in the group whose FCP increased, as well as in the group whose FCP decreased; baseline (1177 mcg/g [IQR 167–2636]), week 8 (53 mcg/g [IQR 0–415]) (*p* < 0.001). Further inspection of the change in the distribution of FCP levels at baseline and week 8 showed that in the cohort whose FCP increased, the majority of participants (8, 62%) had an FCP < 100 mcg/g at baseline, which is consistent with endoscopic healing [20] but at week 8 only one (8%) participant had an FCP < 100 mcg/g (*p* = 0.31). The number of participants with an FCP > 250 mcg/g, which is consistent with active intestinal inflammation, nearly doubled, from five (38%) to nine (69%), from the baseline (*p* = 0.26). The number of participants with an FCP in the range of 100–250 mcg/g, considered a “grey zone” in determining whether active inflammation is present, increased from nought to three (23%) during the study. A similar but opposite trend was seen in the cohort whose FCP decreased; the number of participants with an FCP > 250 mcg/g decreased from nine (56%) to five (31%) (*p* = 0.37), while the number of participants with an FCP < 100 mcg/g increased from three (19%) to nine (56%) (*p* = 0.27). There was no statistically significant difference between the medical therapies used in both cohorts and no therapy change between baseline and week 8 in any participant. Robust linear regression predicted that corticosteroids were marginally associated with the levels of fecal calprotectin at week 8 (0.10 ≤ *p* < 0.05); however, no other medications (e.g., 5-ASA, anti-tumor necrosis factor, immunomodulators, probiotics or antibiotics) were associated with the fecal calprotectin at week 8.

### 2.3. Baseline Microbiome Composition Is Predictive of Fecal Calprotectin Change Trajectory

Principal coordinate analysis (PCoA) based on the taxonomic profile of whole genome sequences demonstrated differences in the baseline microbiome composition between participants whose FCP increased vs. decreased from baseline to week 8 (Figure 2A). While the microbiome composition at baseline was not associated with absolute FCP levels at baseline or week 8, it was predictive of FCP change (increase or decrease) (PERMANOVA *p* < 0.04; R^2^ = 0.047) (Figure 2A). There was a shift in the microbiomes for both cohorts at week 8 (Figure 2B). When the microbiome composition at week 8 was analyzed, it was found to be associated with the absolute FCP levels at week 8 (PERMANOVA *p* < 0.005; R^2^ = 0.057) but not with baseline FCP levels or the change in FCP (Figure 2B). Analysis of the specific microbial species associated with week 8 FCP identified numerous species previously associated with health and inflammation [21], with a few exceptions (Figure 2C). Several species showed increased abundance associated with an absolute decrease in FCP levels, namely the *Roseburia*, *Lachnospiraceae* and *Bifidobacterium* species (spp.). In contrast, several species previously known to be associated with inflammation, including *Bacteroides fragilis, Ruminococcus* spp., and *Eikenella corrodens* [22,23], increased in abundance with an absolute increase in FCP levels. We next examined if there was a link between diet (total macronutrient/micronutrient content (per day), adjusted macronutrient/micronutrient (per 1000 kcal) and food groups) and FCP between participants whose FCP increased vs. decreased from baseline to week 8. There was no significant association of diet with FCP at baseline or week 8 (Appendix A) with the intervention status (Appendix A); however, there was a significant association identified between the change in diet (both adjusted and unadjusted micro- and macronutrients) with the baseline FCP and FCP at week 8 and with the shift in FCP trajectory (Appendix A). Furthermore, a higher MDS score (better adherence) was significantly associated with a reduction in FCP (*p* = 0.004) (Appendix A). Lastly, when we examined MDS, FCP and the microbiome, we identified that the microbes most positively associated with FCP were negatively associated with the change in MDS (Appendix A).

### 2.4. Weighted Gene Co-Expression Network Analysis (WGCNA) 

The WGCNA framework [18,24] was used to examine the two features of interest, intestinal inflammation (as measured by FCP) and diet quality (as measured by the Mediterranean diet score (MDS)), and to identify the feature selection method. The WGCNA was performed on 29 participants who had a change in FCP with matching microbiome profiles at baseline and week-8 time points. There were 551 features identified, based on the following: FCP, microbiome taxa (OTUs), diet profile (total macronutrients and micronutrients, adjusted macronutrients and micronutrients and food groups), serum metabolites, bile acids and short-chain fatty acids.

The 551 features were ranked across all samples using the Spearman rho inter-correlation with the soft-thresholding power set to equal 11 (Appendix A). The module was further refined using a membership score of ≥0.30. There were 189 features that were unclassified (Appendix A). The final network consisted of 10 modules (labelled by color), which contained 362 relevant features ranging in size from 22 to 138 members (Table 2). Notably, the green and turquoise modules were specifically associated with dietary components. While the microbiome taxa were spread across all modules, the grey module was particularly enriched in microbes. The tests of association between the module eigengenes, inflammation (FCP) and diet (MDS) were performed for each module, and the results are summarized in Figure 3A. The green module had a statistically significant positive association with MDS (*p* < 0.05) and a negative association with FCP (*p* < 0.10). The brown and yellow modules both had negative associations with MDS (*p* < 0.10 brown, *p* < 0.05 yellow), and the blue module was positively associated with FCP (*p* < 0.05). There was no significant mediating effect of the turquoise module on the association of MDS with FCP (*p* = 0.78).

When the association networks in the green module were explored further (Figure 3B), the main features of these associations encompassed dietary consumption profiles that included adjusted dietary fiber, frequency of vegetable intake (red/orange, dark green and starchy), legumes, fruit and eggs. These items are also components of the Mediterranean diet. Nutrients included adjusted vitamins (folate, vitamin A, alpha-carotene, lutein, beta-carotene, lycopene, vitamin E and vitamin K), adjusted minerals (iron, magnesium and potassium) and total fatty acids (docosahexaenoic acid and stearidonic acid). The taxa linked to the green module included *Bacteroides eggerthii* and *Bacteroides finegoldii*, which are generally considered a component of a health-associated microbiome, including intake of dietary fiber and pectin fermentation as well as enhancing the efficacy of fecal microbiota transplant (FMT) in IBD [25,26,27].

We next examined the distinct modules enriched for high- and low-risk co-abundant groups (CAGs) (Figure 4). High- and low-risk CAGs were defined based on enterotyping and iterative PAM (partitioning around medoids) clustering of the baseline microbiome. The microbiome was grouped into three main clusters, each marginally associated with baseline FCP. Dominant taxa for the three clusters were identified by selecting only those taxa present in at least 10% of subjects, and abundance was compared using the Kruskal–Wallis test. Only those with an FDR ≤ 0.1 were included. High-risk CAGs included species associated with higher inflammation, while low-risk CAGs included species associated with lower inflammation, based on FCP. The blue, brown and black modules were enriched for high-risk CAG taxa predicted to be positively associated with inflammation. In contrast, the red module was enriched for low-risk CAG taxa predicted to be negatively associated with inflammation. 

Further inspection of the taxa-associated modules revealed that the black module (Figure 5A) was enriched in bacteria from the families Fusobacteriaceae [28], Veillonellaceae, Streptococcaceae [23], Actinomycetaceae and the genus *Rothia.* Many of these bacteria are members of the oral microbiome and are considered in the literature to have pathobiont traits [29,30,31,32]. The brown module, which was negatively associated with the MDS score (Figure 3A), includes microbiome taxa previously associated with IBD and other diseases (Figure 5B), including *Ruminococcus gnavus, Clostridium symbiosum*, *Hungatella hatheway*, *Eggerthella lenta* and *Flavonifractor plautii* [33,34]. Moreover, the brown module indicated a positive correlation value with cholic acid, a primary bile acid previously demonstrated to be elevated in IBD [35]. The serum metabolites 3-2-hydroxyphenyl propanoate and phenylacetaldehyde were also enriched in this module.

The blue module represents the features of SCFAs, serum metabolites, and microbial taxa positively associated with FCP (Figure 5C). The SCFAs (e.g., propionate, isobutyrate and isovalerate), by-products of amino-acid fermentation (e.g., citrulline, L-serine, L-asparagine, L-alanine, L-threonine), bile acid salts (deoxycholate) and taxa including *Streptococcus anginosus*, *Asaccharobacter celatus* and *Enterorhabdus caecimuris* found in this module have been associated with aging [36] and IBD [37].

Finally, the red module, which was associated with low-risk CAGs, was enriched in multiple microbial taxa positively associated with a healthy microbiome, as these microbes have the ability to degrade complex polysaccharides to produce beneficial SCFAs [38] (e.g., *Ruminococcus champanellensis*, *Bacteroides pectinophilus*, *B. xylanisolvens*, *B. ovatus*, *Eubacterium limosum*, and *Akkermensia muciniphila*) (Figure 6). In addition, a positive association with the archaea methanogen *Methanobrevibacter smithii* was found in the red module. 

In summary, through intramodular connectivity (eigengene network analysis), we were able to identify the microbial taxa and metabolites that can mediate the relationship between diet (MDS) and inflammation (FCP).

### 2.5. Inter-Module Association Analysis

Mediation analysis was used to discover the stepwise association pattern that links MDS and FCP to the microbiome-dominated modules (Figure 7). The grey module, which contains all the refined features (bile acids, serum metabolites and microbial taxa) that are positively associated with the MDS and negatively associated with FCP (Table 2), mediates the association between diet and inflammation (Figure 7A, Appendix A). That is, the grey module positively mediates the green module (dietary feature module: a positive association with MDS, negative association with FCP) and negatively mediates the blue module (serum-metabolite-enriched module: positively associated with FCP). The grey module also mediates the relationship between diet (green module) and the low-risk CAG-enriched red module. The yellow module (negatively associated with MDS) is positively associated with both the black module (enriched in disease-associated microbes, specifically several opportunistic pathogens typically found in the oral cavity) and the brown module (pathobiont-enriched module), while the grey module negatively mediates both the yellow and brown modules. Thus, the stepwise association pattern links adherence to the MD and inflammation mediated by microbiome-dominated modules.

Further analyses of the grey module to identify the exact factors associated with the mediation of MDS and inflammation identified 14 features from a total of 65 features (FDR ≤ 0.20 with one feature of MDS and FCP). Figure 7B illustrates the microbial species and metabolites that mediate the relationship between MDS and FCP. All the metabolites, except benzyl alcohol, are phenols for which there are two biologically plausible sources; (1) they are metabolites from polyphenols from the diet or (2) they are metabolites from the microbial metabolism of aromatic amino acids, namely phenylalanine and tyrosine [39]. The MDS is high in polyphenols, which have antioxidant and anti-inflammatory properties [40]. Benzyl alcohol is found in many foods [41] and is also used as a preservative in medications [42]. All the bacterial species identified mediate the relationship between MDS and FCP through various pathways, except for *Roseburia faecis* and *Fuscatenibacter saccharivorans*, which are directly associated with FCP.

Finally, we used mediation modelling to evaluate the indirect effects mediating MDS and FCP. We identified that a specific health-associated microbiome guild of four taxa (*Faecalibacterium prausntizii, Eubacterium ramulus, Dorea longicatena* and *Roseburia inulinivorans*) significantly influence the association between MDS and FCP in a three-step manner (Figure 8A). The positive association between MDS and *Dorea longicatena* is mediated by *Faecalibacterium prausntizii* (*p* = 0.02) and *Eubacterium ramulus* (not statistically significant*, p* = 0.06), while the negative association between *D. longicatena* and FCP is mediated by *Roseburia inulinivorans* (*p* = 0.002). There was also a metabolite cluster mediating the negative association between MDS and FCP. Both benzyl alcohol (ACME −0.66, *p* = 0.04) and 3-hydroxyphenylacetate (ACME = −0.84, *p* = 0.03) were found to mediate the negative association between MDS and FCP. There was a positive association between MDS and 3-4-hydroxyphenylacetate (ACME 4.32, *p* = 0.03), which then positively mediated the association between MDS and phenyl acetate; phenyl acetate, in turn, increased the negative association between MDS and FCP (ACME −0.06, *p* = 0.04) (Figure 8B). Finally, when combined, the selected group of features (three taxa and four metabolites) had an even stronger mediating effect between MDS and FCP (ACME –1.23; *p* = 0.004) (Figure 8C).

## 3. Discussion

The links between the Mediterranean diet (MD) and the microbiome and host responses are gradually becoming known [14,21,43]. This is the first study, to our knowledge, to explore in detail the influence of the MD on gut microbial taxa, bile acids, SCFAs and metabolites and their relationship with intestinal inflammation (FCP) in individuals with UC. Through WCGNA, we: (1) identified distinct modules with specific diet-associated features and/or metabolites associated with MDS and FCP and (2) demonstrated that microbiome-dominated modules mediated stepwise association patterns linking the MDS and inflammation-associated modules. Finally, through mediation analysis, we identified a health-associated guild of three taxa of bacteria and four metabolites which had a strong mediating effect between MDS and FCP.

The MD is characterized by the enriched intake of fruits, vegetables, nuts, and legumes with fewer red meats and refined grains [44] and has been associated with reduced risk of several chronic and inflammatory diseases [36,45,46]. A deeper inspection of the high-risk CAG-enriched modules (microbial taxa associated with increased FCP) yielded some interesting observations and possible mechanisms in the relationship of these CAGs to diet and IBD. For example, the black module was enriched largely by bacterial species, which are members of the oral microbiota, including species of *Streptococcus, Veillonella, Fusobacteria* and *Actinomyces* [47]. There is growing evidence of a link between oral dysbiosis and disease, namely periodontitis and IBD [48,49], with many oral commensals detected in the lower GI tract in IBD [48]. Moreover, many of these bacteria are opportunistic pathogens and sugar-rich diets fuel oral dysbiosis and the emergence of pathogens [50,51]. Interestingly, the black module showed a trend towards a negative association with MDS and a positive association with FCP but did not reach statistical significance. Thus, it is plausible that diets high in sugar, which would therefore have a low MDS, might drive oral dysbiosis, leading to an abundance of these oral opportunistic pathogens in the lower GI tract in IBD, where they may contribute to intestinal dysbiosis and inflammation. 

The brown module was negatively associated with MDS and FCP; however, only the association with MDS was statistically significant (Figure 2A). Interestingly, while this module was enriched by a large number of pathobionts, it also contained some health-associated bacteria, such as *Lactobacillus fermentum* and *Bifidobacterium gallinorum*. Several species of *Blautia* were also present and have been associated with fast-food consumption [52], which could contribute to this module’s negative association with MDS. Interestingly, previous studies have shown mixed results regarding the role of *Blautia* spp. in IBD; some studies have found it to be increased in IBD [53], while others have found it to be decreased [54] and there is evidence that its ability to produce bacteriocins may be beneficial in gut health and dysbiosis [55]. It is possible that the effect of *Blautia* spp. on intestinal inflammation and health may be dependent on host-specific factors, such as disease severity, genetics or relative abundance of other related bacterial species in the host microbiome, and thus highlights the difficulty in understanding the complex relationships between diet, the microbiome and the metabolome in health and disease.

Using mediation analysis, we identified specific microbiome and metabolic features which mediate the relationship between MDS and FCP (Figure 6B). Four of the five metabolites identified were phenols, which are metabolites of polyphenols found in plants and considered “candidate” prebiotics. The Mediterranean diet is high in polyphenols, which, in addition to having antioxidant and anti-inflammatory properties, may also contribute to intestinal barrier function, modulation of immune signaling and inhibition of pathogenic bacteria and pro-inflammatory mediators [56]. Our findings are supported by a previous study which found an association between adherence to the Mediterranean diet (MDS) and increased levels of phenols, specifically phenyl acetate and 3-hydroxyphenylacetate [57]. Similarly, De Filippis et al. (2016) [14] found a positive association between plant-based diets (rich in intrinsic fibers) and increased *Roseburia* spp. and SCFAs. SCFAs are known to be beneficial to intestinal health and associated with reduced FCP [58]. A synergistic anti-inflammatory effect between phenolic metabolites and SCFAs has been demonstrated in vitro [59]. Many of the bacteria found to mediate the negative association between MDS and FCP (Figure 7B) in our study (*Faecalbacterium prausnitzii, Roseburia* spp., and *Ruminococcus* spp.) possess the enzymes required to metabolize aromatic amino acids, namely phenylalanine and tyrosine, to phenolic compounds [60]. These compounds have various physiological effects, including immunomodulatory functions, thus suggesting another possible mechanism by which these bacteria mediate the relationship between MDS and FCP [60]. *F. prausnitzii*, *Roseburia* spp., and *Ruminococcus* spp. all possess enzymes required to convert phenylalanine to phenylpyruvate, an intermediate metabolite that can then be converted to phenylacetate by common non-pathogenic *E. coli* [61]. *Ruminococcus* spp. can metabolize tyrosine to 3—4 hydroxyphenyl pyruvate, which can then be further metabolized to produce other phenolic compounds, including neurotransmitters in the body [61]. *Eubacterium ramulus* mediated the association between MDS and *Dorea longicatena* but did not meet significance (Figure 8A). Both *E. ramulus* and *Dorea longicatena* possess flavonoid-modifying enzymes [62,63]. Flavonoids are a major class of plant polyphenols [64], and thus it is possible that flavonoids found in the MD play a role in the relationship between these bacteria.

Whole food, plant-based diet interventions, such as the MD are interesting and represent how humans normally eat, as humans do not consume fiber as an isolated component. Vegetables, fruit, nuts, seeds, legumes, and grains are whole foods that are not just one single source or extract of fiber but contain a complex three-dimensional plant cell matrix (i.e., plant cell walls), termed “intrinsic fibers” [65]. Various types of fiber are stored in vacuoles (e.g., starch, fructans, sugars, and phytochemicals) within the plant cell walls, which differ according to the plant source. As a result, the three-dimensional plant cell matrix has important consequences for the microbiota in accessing the individual fibers, influencing digestion and fermentation patterns [65]. Although there is a paucity of human clinical trials that have examined the association of intrinsic fibers and the gut microbiome, increased levels of health-associated microbial taxa involved in the breakdown of fiber, are observed [65]. In the current study, the three main bacteria taxa identified in the health-associated microbiome guild (Figure 8A) are known to metabolize a variety of fibers; *F. prausnitzii* and *R. inulinivorans* ferment β-fructans, including inulin, along with galactooligosaccharides and xylooligosaccharides, while *D. longicatena* is stimulated by short-chain fructooligosaccharides [66,67,68]. In summary, a plant-based whole-food diet such as the MD can influence the diversity, activity and functionality of various gut microbes, fostering the production of healthy metabolites that can confer multiple benefits on the host’s health.

We highlight some interesting observations from the clinical trial. Despite a significant decrease in sulfur intake in the MD intervention from baseline to week 8, this did not translate into reduced FCP. We noted a higher concentration of valeric acid in the MD-intervention group versus the control group. Preliminary research confirms valerate’s beneficial role in reducing mucosal inflammation, with healthy individuals producing higher levels of valerate, which is associated with a health-associated microbiome [69,70]. The specific mechanisms of action need to be elucidated; however, some hypothesize that dietary changes that restore healthy gut microbiota and raise valeric acid levels will benefit patients with IBD [69]. Finally, we observed a decrease in the conjugated bile acid, glycochenodeoxycholic acid, in the low-sulfur MD-intervention group versus the control group. Disturbances in bile acid metabolism are associated with IBD, as they play an important role in perpetuating the pro-inflammatory cycle between microbiota and the host [35,71,72]. Specifically, fecal-conjugated bile acids are significantly higher in active IBD [35]. Looking at our results, we suspect that the lower levels of glycochenodeoxycholic acid found in the MD intervention could be related to the lower overall clinical disease activity (e.g., PMS, FCP) observed in the MD intervention at baseline.

The clinical study sample was heterogeneous at baseline regarding disease activity, as indicated by the wide IQR for PMS and total Mayo score. Additionally, active disease was managed by standard medical care, which may have diluted the diet response. Regarding the diet intervention, there was no difference in baseline MDS between intervention groups and no change in MDS between time points in the intervention group. Thus, this highlights the importance of assigning participants to a dietary-intervention group in which the diet intervention will be measurably different from the participants’ baseline diet. Behavioral change is difficult, and therefore frequent check-ins and weekly adherence checks need to be routinely implemented into dietary clinical trials to ensure high compliance with the diet regimen.

The clinical study and post hoc analysis have several strengths. To our knowledge, this is the first study that has used WGCNA to examine the mediation effects between inflammation, the gut microbiome and diet in IBD. Most studies to date include one type of -omics data, which provides only limited insights into the biological mechanisms of disease [73]. Human diseases involve complex and dynamic networks performing biological functions; therefore, integrative data analysis methods can only capture the complementary effects and synergistic interactions between -omics data [73]. In addition, our -omics dataset was collected on the same set of samples, providing further rigor to our findings. Despite the advantages, the study has some limitations, mainly its small sample size and the heterogeneity of the participants, which limit the statistical power to detect other findings; nevertheless, we can use the current findings to generate scientific hypotheses that we can formally study in future randomized studies. Specifically, is there a role for microbiome testing to determine the presence of a functional guild in patients with the metabolic capacity to have a favorable response to dietary interventions? Is there a way to modify the microbiome to produce an ‘optimal’ functional guild through pre- or probiotics or fecal microbiota transplant? Can we identify the foods within the Mediterranean diet from which health-associated metabolites directly or indirectly inhibit intestinal inflammation, or can we exploit these metabolites for molecule development to generate new therapeutic opportunities in IBD?

## 4. Materials and Methods

### 4.1. Study Design and Participants

The present study is a post hoc analysis based on data obtained from an open-label, 8-week randomized controlled clinical trial (*n* = 40) designed to examine how modulating dietary sulfur intake with a Mediterranean-like diet pattern impacts markers of inflammation, the gut microbiome, SCFAs, bile acids and metabolome in UC. Following informed consent, participants were randomly allocated to either the control or intervention groups. Randomization was performed by a computerized random number generator, and group assignment was concealed from the study coordinator enrolling patients until the time of assignment using sealed envelopes. Appendix A shows the Consort study flow diagram.

The intervention group received diet counselling from an RD, which included a Mediterranean-like diet pattern (MD), reducing sulfur-rich foods, beverages, food additives (e.g., carrageenan), reducing beverages high in sulfate/sulfur and reducing sulfur-containing supplements (e.g., chondroitin sulfate). The control group was asked to make no changes to their diet (habitual diet) and received one session with an RD on how to eat a Mediterranean-like diet at the end of study participation (week 8). Both groups received conventional treatment with visits to the clinic at four weeks to ensure completion of their 24-h food recalls and food-frequency questionnaires.

The study was conducted in the outpatient IBD clinic at the University of Calgary from April 2016 to January 2021. The inclusion criteria for the intervention trial were adult patients (age > 18 years) with UC established by usual endoscopic and histologic criteria. Patients experiencing a UC flare or in clinical remission were eligible. Those with a clinical disease flare, defined as partial Mayo score (PMS > 2), were managed by their gastroenterologist with any combination of 5-ASA (oral, topical or both), an immunomodulator, oral corticosteroids or the initiation of a new biologic as part of their usual care. Participants with UC in clinical remission were defined by PMS ≤ 2 and maintained on any combination of the therapies mentioned above. Participants were required to have a baseline endoscopic assessment (flexible sigmoidoscopy or colonoscopy) and strongly encouraged to have an endoscopic reassessment at the end of the study. The exclusion criteria included major medical comorbidities (diabetes, active malignancy within the past five years, active infections, severe respiratory or cardiac disease, acute or chronic kidney disease), previous bowel surgery or smoking. The protocol deviations are reported in the Appendix A. No adverse events were reported.

### 4.2. Assessment of Diet Intake

Food and beverage intake reported by the participants were assessed at baseline and week 8 using two non-consecutive 24-h food recalls using the Automated Self-Administered 24-h (ASA-24^®^) Dietary Assessment Tool (Canadian version) [74]. Details of each food and beverage item, food groups and nutrient intakes for macro- and micronutrients were downloaded from the ASA-24 researcher website. Guidelines from the US National Cancer Institute were applied for data cleaning [75]. Macro- and micronutrients were also evaluated in relation to energy intake (per 1000 kilocalories), referred to as adjusted macro- and micronutrients. ASA-24 does not capture sulfur or sulphate intake; therefore, mean values were calculated for each participant using values described in the literature [76,77]. The English version of the *PREvencion con DIetaMEDiterranea* 14-item Mediterranean Diet Adherence Screener was used to calculate a Mediterranean diet score (MDS) from the diet records with one modification [78]. As alcohol consumption is contraindicated in IBD, points were not provided for consuming alcohol [79].

### 4.3. Fecal Microbiome Analysis

Microbiome analysis was performed at the International Microbiome Centre (IMC) at the University of Calgary (Calgary, Canada). Details of DNA isolation are described in the Appendix A. Briefly, the raw FASTQ files were first filtered for adapter sequences and end trimmed of bases with a quality of less than 15. Quality-controlled reads were analyzed using taxonomic-based approaches such as GAST and the Ribosomal Database Project MultiClassifier tool and non-taxonomic-based clustering algorithms for Operational Taxonomic Unit determination with the UPARSE pipeline. Alpha-diversity (observed species, Shannon, Simpson) and β-diversity indices (Bray–Curtis, binary Jaccard) were calculated in QIIME II and R (VEGAN package). Ordination plots for β-diversity metrics were generated by non-parametric multidimensional scaling ordination in R. To assess the microbiome’s functional composition, the microbial community’s gene content was inferred using the PICRUSt algorithm. OTUs tables generated with QIIME II/UPARSE were normalized by 16S rRNA gene copy number and the normalized values multiplied by the calculated abundance of gene families in each taxon during the gene content inference procedure performed with PICRUSt. The result is a table of gene family counts that is comparable to those generated by metagenome annotation pipelines such as HUMAnN and MG-RAST, and which can be organized into metabolic pathways.

### 4.4. Serum Metabolome Analysis

Fecal samples were extracted according to the previously published methods [80]. Briefly, samples were extracted with ice-cold 50% methanol in a 5:1 (*v*/*w*) ratio (µL/mg). Samples were homogenized at 30 Hz for 3 min on a homogenizer (Bead Ruptor 96, OMNI International, Kennesaw, GA, USA), incubated for 30 min on ice and then centrifuged for 10 min at 10,000× *g*. A total of 20 µL of debris-free supernatant was diluted 10 times in 50% methanol and analyzed using liquid chromatography–mass spectrometry (LC-MS).

Metabolic analysis was performed according to previously published studies [81,82,83]. Briefly, chromatographical separation of metabolites was performed on a Syncronis HILIC UHPLC column (2.1 mm × 100 mm × 1.7 μm, Thermo-Fisher, Burnaby, BC, Canada) at the flow rate of 600 μL/min using a binary solvent system: solvent A, 20 mM ammonium formate pH 3.0 in mass spectrometry grade H_2_0 and solvent B, mass spectrometry grade acetonitrile with 0.1% formic acid (% *v*/*v*). The following gradient was used: 0–2 min, 100% B; 2–7 min, 100–80% B; 7–10 min, 80–5% B; 10–12 min, 5% B; 12–13 min, 5–100% B; and 13–15 min, 100% B. The sample injection volume was 2 μL. The masses were acquired on a Q Exactive™ HF Hybrid Quadrupole-Orbitrap™ Mass Spectrometer (Thermo-Fisher, Canada) coupled to a Vanquish™ UHPLC System (Thermo-Fisher, Canada). The mass spectrometer was run in negative full-scan mode at a resolution of 240,000 scanning from 50 to 750 *m/z*.

Metabolite data was analyzed using the El-MAVEN software package [84,85]. Metabolites were identified by matching observed *m/z* signals (±10 ppm) and chromatographic retention times to those observed from commercial metabolite standards (LMSLS^TM^ Sigma-Aldrich, Oakville, ON, Canada).

### 4.5. Bile Acids

Fecal samples were extracted with ice-cold methanol in a 2:1 (*v/w*) ratio (µL/mg). Samples were homogenized at 30 Hz for 3 min on a homogenizer (Bead Ruptor 96, OMNI international, Kennesaw, GA, USA) equipped with microcentrifuge tube tissuelyser adapter sets (2 × 24, Qiagen, Germantown, MD, USA). After a first round of centrifugation at 18,000× *g* for 10 min at 4 °C, supernatants were collected and submitted to another centrifugation step under the same conditions. Clear supernatants were diluted 1:2 with 50% MeOH (HPLC-grade) before LC-MS/MS analysis.

LC-MS/MS analysis was performed on a Vanquish^TM^ ultra-high-performance liquid chromatography (UHPLC) system coupled to a TSQ Quantum^TM^ Access MAX triple quadrupole mass spectrometer (Thermo Fisher Scientific, Burnaby, BC, Canada) equipped with an electrospray ionization (HESI-II) probe. The UHPLC-MS platform was controlled by an Xcalibur^TM^ data system (Thermo Fisher Scientific, Burnaby, BC, Canada). Chromatographic separation was achieved on a Hypersil GOLD ^TM^ C18 column (200 mm × 2.1 mm, 1.9 µm, Thermo Fisher Scientific, Burnaby, BC, Canada) using a binary solvent system composed of LC-MS grade H_2_O containing 5 mM ammonium acetate and 0.1% (% *v/v*) formic acid (Solvent A) and LC-MS grade MeOH containing 5 mM ammonium acetate and 0.1% (% *v/v*) formic acid (solvent B). The following 40 min gradient, with a chromatographic resolution of monitored isobaric bile acids (chenodeoxycholic acid/deoxycholic acid/hyodeoxycholic acid/ursodeoxycholic acid and taurochenodeoxycholic acid/taurodeoxycholic acid, was used: 0–5 min, 60% B; 5–12 min, 60–80% B; 12–20 min, 80% B; 20–24 min, 80–100% B; 24–34 min, 100% B, 34–35 min, 100–60% B, and 35–40 min, 60% B. The flow rate was 250 µL min^−1^, and the sample injection volume 5 µL. The autosampler was kept at 6 °C and the column at 30 °C.

MS/MS data were acquired in positive electrospray ionization mode with the mass spectrometer operating in selected reaction monitoring (SRM) mode. Fragmentation parameters were optimized using the EZ Tune program with direct infusion of the analytical grade bile acid standards (50 µM each in 50% solvent A–solvent B). For each bile acid, a pair of quantifier (Quan) and qualifier (Qual) ions was selected and monitored, with a scan time of 0.05 sec: Bile acid, [M + NH4]^+^ *m*/*z* Parent ion → *m*/*z* Quan (CE), *m*/*z* Qual (CE), where CE is the collision energy (V). Further details are provided in the Appendix A.

Electrospray ionization source conditions were as follows: spray voltage of 3000 V, vaporizer temperature of 200 °C, sheath gas of 20 psi, auxiliary gas flow of 2 (arbitrary units), sweep gas flow of 1 (arbitrary units), and capillary temperature of 200 °C.

Bile acids in the biological samples were quantified using an external standard diluted at four different concentrations to construct the calibration curve (Appendix A). Data analyses on the converted mzXML files were conducted in El-MAVEN using integrated peak intensity (area under the curve) [31]. Reported bile acid concentrations in the samples were corrected for sample dilutions (1:2) before LC-MS/MS analysis.

### 4.6. Short-Chain Fatty Acids

SCFAs were quantified according to previous methods [86]. In brief, SCFAs were extracted (1:2 ratio of wet sample weight (mg) to extraction solvent (µL)) from fecal samples with ice-cold extraction solvent (50% water/acetonitrile, *v*/*v*) spiked with stable isotope-labeled internal standards (IS) (acetic acid-1,2-13C_2_, 4 mM, final concentration; propionic acid-13C_3_, 1 mM; butyric acid-1,2-13C_2_, 1 mM; isobutyric acid-d7, 250 µM, valeric acid-d9, 500 µM; and isovaleric acid-d9, 250 µM), homogenized at 30 Hz for 3 min with a tissue lyser (Qiagen, Germantown, MD, USA), derivatized with N-(3-Dimethylaminopropyl)-N′-ethylcarbodiimide hydrochloride (EDC) and aniline, then diluted (1:10) with H2O/MeOH (50:50, *v/v*) prior to LC-MS/MS analysis. LC-MS/MS analysis and data analyses were performed according to Bihan et al. (2022) [86].

### 4.7. Weighted Gene Co-Expression Analysis (WGCNA)

Correlation analysis of association among features in our data types (species level microbiome, fecal metabolome, fecal SFCAs, fecal bile acids and diet variables) were performed using WGCNA. It is a very popular method used for computing gene co-expression networks which have recently been used to integrate multi-omics. To create WGCNA networks, we used a soft thresholding power range from 1 to 20. The scale-free topology model fit (R^2) reached 0.42 when the soft thresholding power was 10. We used the minimum module size of 11. The features were clustered into modules by hierarchical clustering with Euclidian distance. Module membership was assigned using the signedKME function and the most similar modules were merged by setting the MEDissThres (module eigen dissimilarity threshold) to 0.25.

### 4.8. Statistical Methods

The Anderson–Darling test was used to assess the normality of the data. Continuous variables are presented as means and standard deviation (SD), or median and interquartile range (IQR), and categorical data are presented as absolute value and percentage. The Wilcoxon matched-pairs signed-rank test (non-parametric) and paired *t*-test (parametric) were used for paired data, and the Mann–Whitney U test (non-parametric) or unpaired (*t*-test) was used for the comparison between groups. Categorical variables were compared using Fisher’s exact test. A permutational multivariate analysis of variance (PERMANOVA) was used to examine distance matrices. *p*-values of <0.05 were considered statistically significant. The statistical package GraphPad Prism Version 9.3.0 (Graph Pad Software, San Diego, CA, USA) and R [87] were used for the analyses and figures.

### 4.9. Power Calculation

The open-label clinical trial was powered to detect a clinically significant mean difference of 3 points on the PMS between and within groups at power = 0.90 and alpha = 0.05 for an estimated sample size of 21 per group.

## 5. Conclusions

We identify microbial taxa and metabolites associated with the MD that mediate the association between the MD and FCP. These novel findings provide an important insight into how the MD may influence the clinical course in IBD. It is thus plausible to suggest that patients with IBD should be encouraged to optimize their diet through increased consumption of foods, including olive oil (four tbsp/day), vegetables (four servings/day), and fruits (three servings/day), which are typical of the MD because they are rich in polyphenols and fiber, leading to beneficial outcomes in IBD.

## Figures and Tables

**Figure 1 ijms-24-07323-f001:**
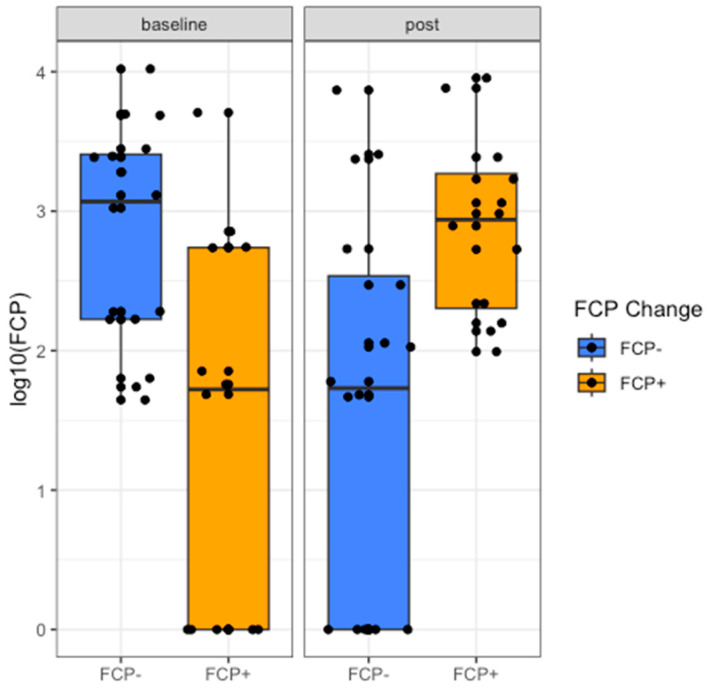
The identification of two distinct trajectories for fecal calprotectin. Participants whose fecal calprotectin increased at week 8 had significantly lower levels at baseline and significantly higher levels at week 8.

**Figure 2 ijms-24-07323-f002:**
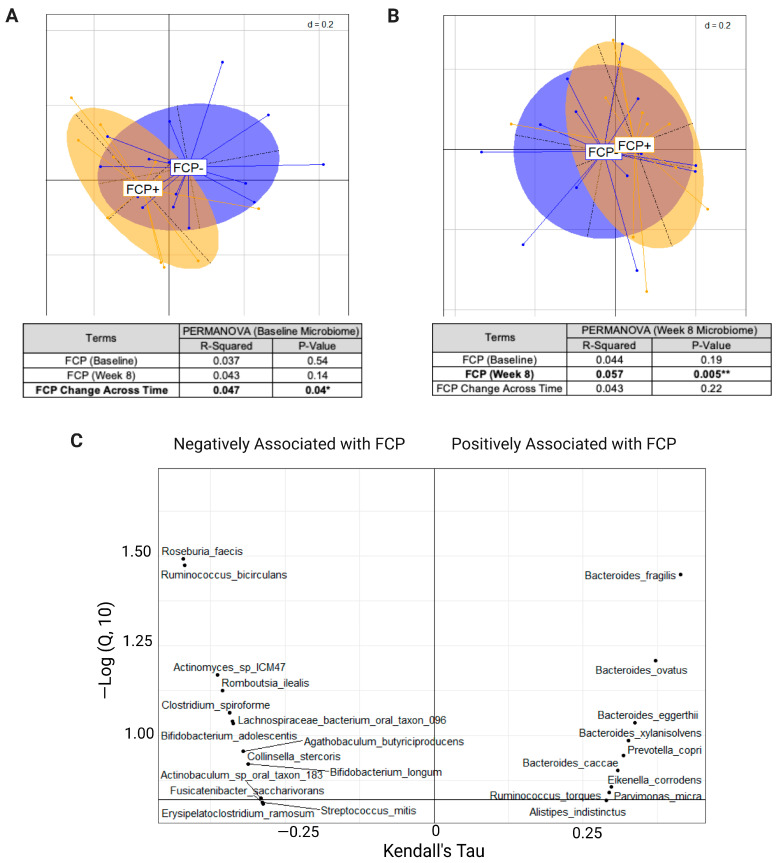
Associations between fecal calprotectin (FCP) and the microbial taxa. (**A**) PCoA plot of baseline microbiome profiles of participants that had an increase in FCP (FCP+; orange) vs. a decrease in FCP (FCP−; blue) from baseline to week 8. PERMANOVA values of the association of baseline microbiome with baseline FCP, week 8 FCP and change in FCP are indicated below the figure. (**B**) PCoA plot of week 8 microbiome profiles of participants that had an increase in FCP (FCP+; orange) vs. decrease in FCP (FCP−; blue) from baseline to week 8. PERMANOVA values of the association of week 8 microbiome with baseline, week 8 FCP and change in FCP are indicated below the figure. (**C**) Identification of the specific species of microbes associated with fecal calprotectin at week 8. Asterisks indicate the significance level: ** *p* < 0.01, * *p* < 0.05. Abbreviations: FCP, fecal calprotectin; PCoA, principle coordinate analysis; PERMANOVA, permutational multivariate analysis of variance.

**Figure 3 ijms-24-07323-f003:**
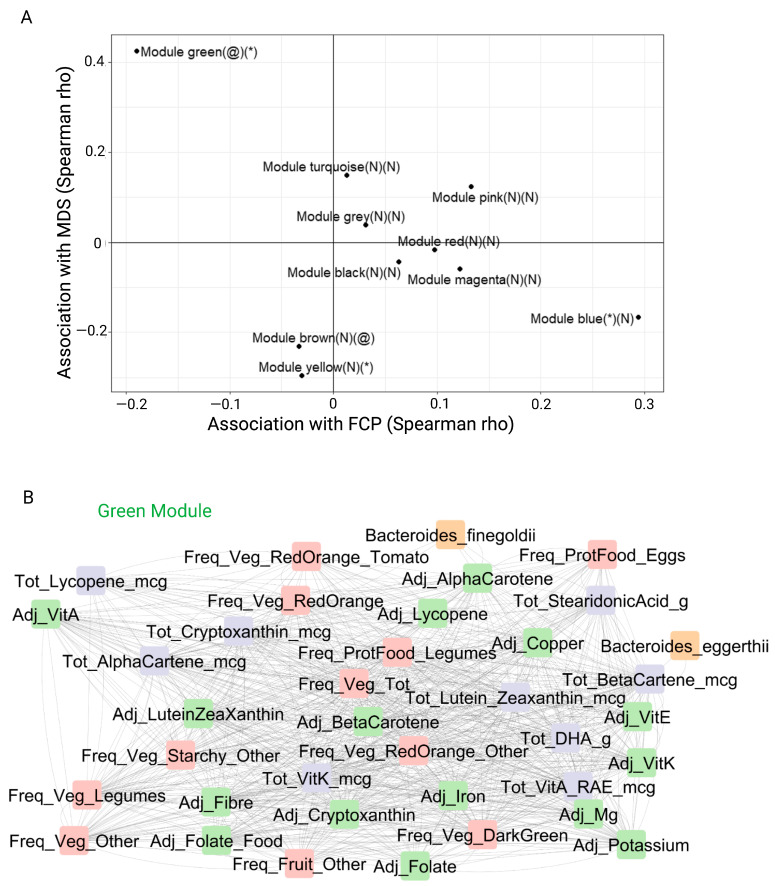
Associating module eigengenes with fecal calprotectin and Mediterranean diet score. (**A**) The Spearman’s rho inter-correlation between cluster eigengenes. Significance level: * *p* < 0.05, @ *p* < 0.10, N = not significant. (**B**) The negative association with fecal calprotectin (green module) encompasses specific nutrients, foods and *Bacteroides* species. Freq refers to the frequency of food groups (shown in pink); Tot refers to the macronutrient and micronutrients calculated per day (shown in purple); Adj refers to macronutrients and micronutrients calculated per 1000 kilocalories (shown in green); Bacteroides species (shown in orange). Abbreviations: MDS, Mediterranean diet score; FCP, fecal calprotectin; Veg, vegetables; Mg, magnesium; DHA; docosahexaenoic acid, Vit, vitamin; Prot, protein; mcg, micrograms; g, grams; RAE, retinol activity equivalents; BetaCartene, beta-carotene.

**Figure 4 ijms-24-07323-f004:**
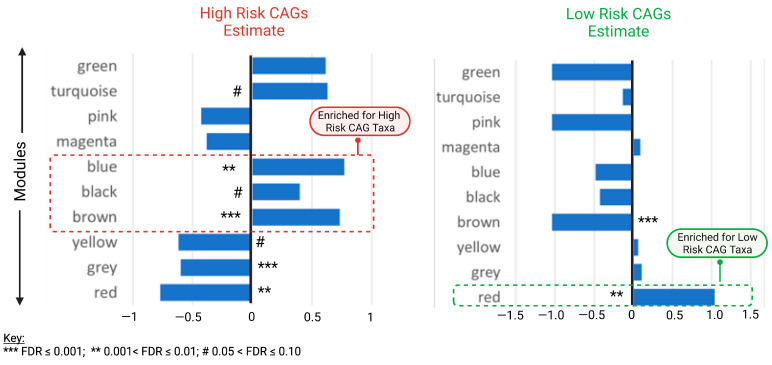
Discrete modules for low- and high-risk Co-Abundant Groups (CAGs). FDR, false discovery rates.

**Figure 5 ijms-24-07323-f005:**
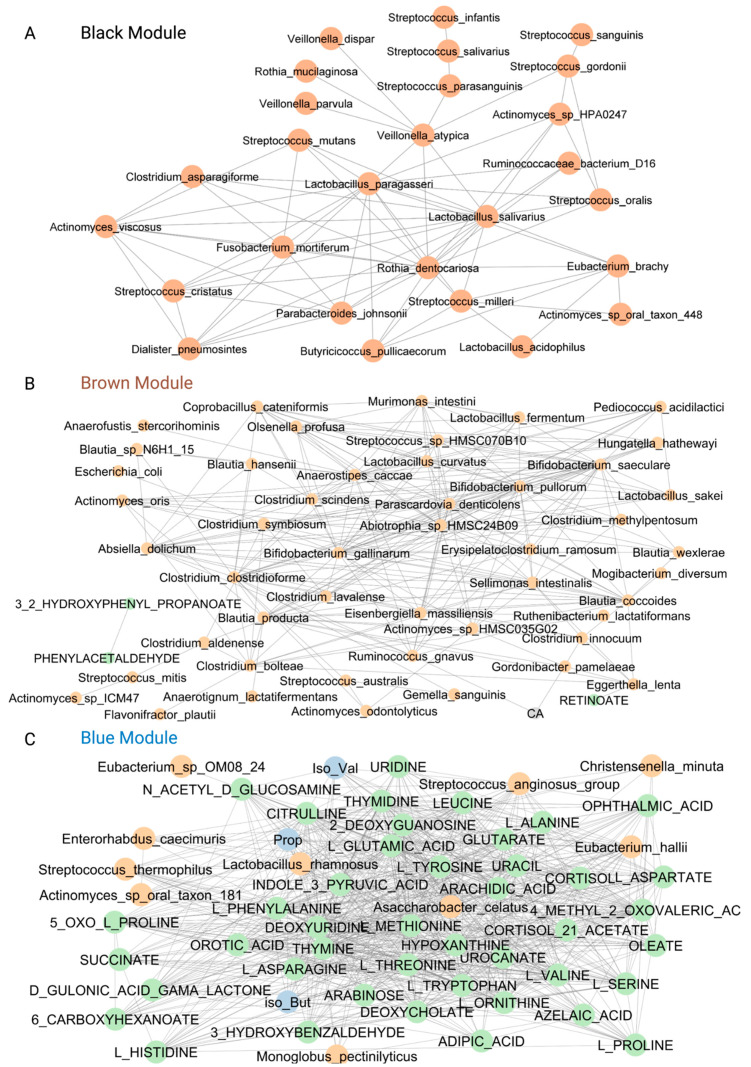
Modules enriched for High-Risk CAG taxa. (**A**) The black module shows taxa (shown in orange) associated with the three clusters predicted to have pathobiont traits. Abbreviations: sp, species. (**B**) The brown module shows species with pathobiont characteristics that are known to be increased in disease. Abbreviations: CA, cholic acid; AC, acid. (**C**) The blue module shows microbial taxa and metabolites (SCFA shown in blue, all other metabolites shown in green) often elevated in disease. Abbreviations: Iso_val, isovalerate; iso_But, isobutyrate; pro, propionate; 5-OXO-L proline, 5-oxoproline (pyroglutamic acid).

**Figure 6 ijms-24-07323-f006:**
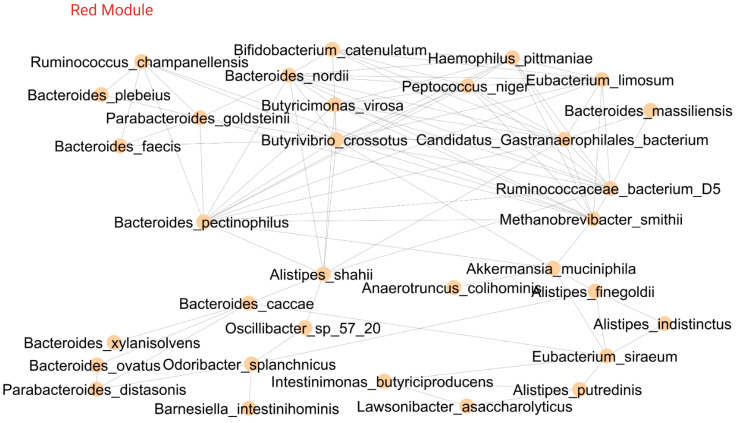
Low-risk CAG module (red module) with health-associated microbes.

**Figure 7 ijms-24-07323-f007:**
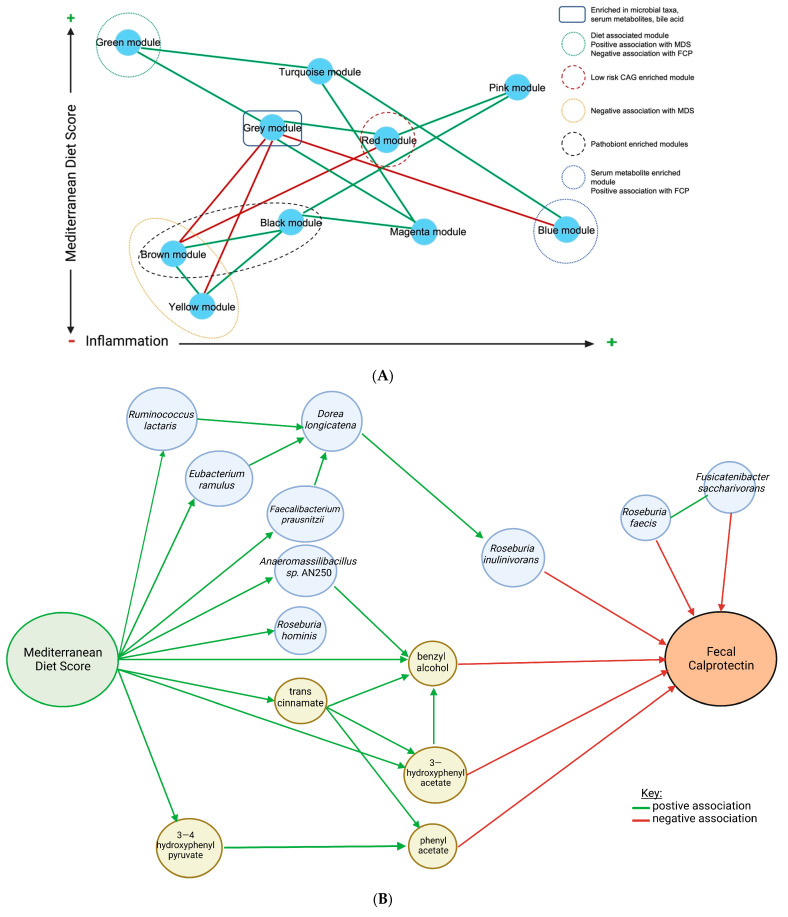
Inter-module association analysis. (**A**) Stepwise association pattern linking high adherence to the Mediterranean diet and fecal calprotectin mediated by microbiome-dominated modules. (**B**) Analysis of the sub-network amongst the select features from the grey module indicates specific microbiome and metabolite features that act as mediators in the relationship between fecal calprotectin and the Mediterranean diet score. Abbreviations: MDS, Mediterranean diet score, FCP, fecal calprotectin. Created with BioRender.com (accessed on 5 January 2023).

**Figure 8 ijms-24-07323-f008:**
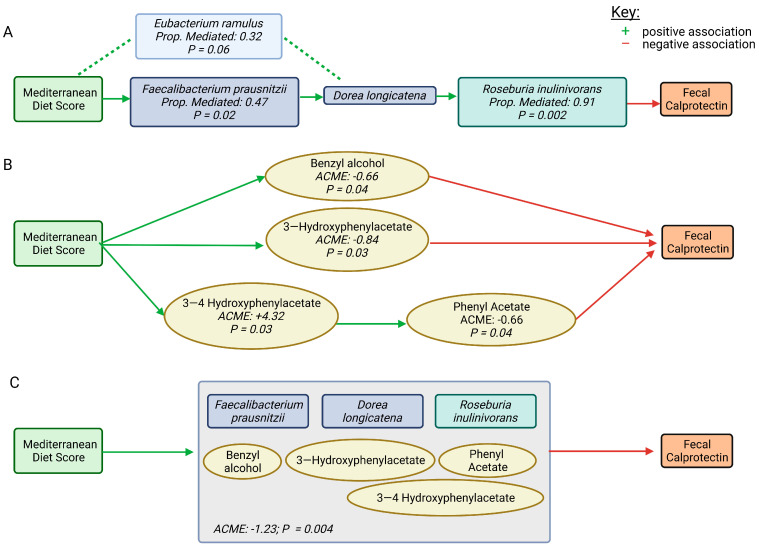
Weighted gene co-expression network analysis (WGCNA). (**A**) A specific health-associated microbiome guild of four taxa significantly influences the association between Mediterranean diet score and fecal calprotectin in a 3-step manner. (**B**) The metabolite cluster mediates the negative association between the Mediterranean diet score and fecal calprotectin. (**C**) A combination of features (three taxa and four metabolites) demonstrates an even stronger mediating effect between the Mediterranean diet score and fecal calprotectin. Green arrows represent a positive association and red arrows represent a negative association. Abbreviations: MDS, Mediterranean diet score; FCP, fecal calprotectin; ACME, average causal mediation effect; prop. mediated; proportion mediated. Created with BioRender.com (accessed on 5 January 2023).

**Table 1 ijms-24-07323-t001:** Characteristics of the study participants with changes in fecal calprotectin from baseline to week 8 (*n* = 29).

	Cohort with an Increase in Fecal Calprotectin at Week 8 (*n* = 13) Median (IQR)			Cohort with a Decrease in Fecal Calprotectin at Week 8 (*n* = 16) Median (IQR)			*p*-Value	
** *Patient Characteristics* **								
**Male, *n* (%)**	4 (31%)			11 (69%)			0.07 ^#^	
**Age (years)**	39 (31.0–47.0)			31 (27.0–43.5)			0.25 ^a^	
**Body mass index** **(Kg/m^2^)**	25.2 (24.3–27.3) (*n* = 12)			25.4 (23.8–28.1) (*n* = 14)			0.72 ^a^	
**Total Mayo Score**	6.0 (4.0–7.0) (*n* = 9)			6.5 (5.0–7.0) (*n* = 10)			0.77 ^a^	
**Partial Mayo Score**	2 (1–4)			3 (1–5) (*n* = 15)			0.73 ^a^	
** *Partial Mayo Score by disease activity* **								
**Remission, *n* (%)**	6 (46%)			4 (27%)			0.55 ^^^	
**Mild, *n* (%)**	4 (31%)			7 (47%)			0.60 ^^^	
**Moderate, *n* (%)**	2 (15%)			2 (13%)			0.95 ^^^	
**Severe, *n* (%)**	1 (8%)			2 (13%)			n/a ^^^	
** *Medical Therapy* **								
**5-ASA monotherapy** ***n* (%)**	5 (38%)			7 (44%)			0.24 ^#^	
**Anti-TNF** ***n* (%)**	3 (23%)			4 (25%)			1.00 ^#^	
**Corticosteroid** ***n* (%) ***	3 (25%) (*n* = 12)			6 (38%)			0.69 ^#^	
**Immunomodulator, *n* (%) ***	1 (8%)			2 (13%)			1.00 ^#^	
**Probiotics** ***n* (%) ***	4 (31%)			3 (19%)			0.67 ^#^	
** *Fecal Calprotectin (FCP)* **								
	Baseline	Week 8	*p*-value	Baseline	Week 8	*p*-value	Baseline	Week 8
**FCP (mcg/g)**	56.3 (0–551)	961 (217–2436)	0.001 ^b^	1177 (167–2636)	53 (0–415)	<0.001 ^b^	0.02 ^a^	0.01 ^a^
**FCP < 100, remission** ***n* (%) ^†,^^**	8 (62%)	1 (8%)	0.31	3 (19%)	9 (56%)	0.27	0.20	n/a
**FCP 100–250,** **grey zone** ***n* (%) ^†,^^**	0	3 (23%)	n/a	4 (25%)	2 (13%)	0.74	n/a	0.78
**FCP > 250,** **active** ***n* (%) ^†,^^**	5 (38%)	9 (69%)	0.2	9 (56%)	5 (31%)	0.37	0.52	0.17

Abbreviations: 5-ASA:5-aminosalicylic acids; Anti-TNF: anti-tumor necrosis factor. * used concomitantly with other therapies, ^†^ based on the International Organization for the Study of IBD guidelines [20] ^#^ Fisher’s exact test; ^^^ test of proportions, n/a = insufficient counts; ^a^ Mann–Whitney test; ^b^ Wilcoxon signed-rank test.

**Table 2 ijms-24-07323-t002:** Differential association of modules with specific feature types.

Modules	Macronutrients and Micronutrients (Total) ^a^	Macronutrients and Micronutrients (Adjusted) ^b^	Food Groups (Frequency) ^c^	Microbiome Taxa	Serum Metabolites	SCFA	Bile Acids
black	0	0	0	28	0	0	0
blue	0	0	0	16	48	5	0
brown	0	0	0	55	3	0	1
green	9	15	11	2	0	0	0
grey	0	0	0	113	22	0	3
magenta	0	0	0	21	0	0	1
pink	0	0	0	23	0	0	0
red	0	0	0	32	0	0	0
turquoise	38	22	26	8	6	1	0
yellow	0	0	0	33	5	0	4

^a^ Macronutrients and micronutrients calculated per day. ^b^ Macronutrient/micronutrients calculated per 1000 kilocalories. ^c^ Frequency refers to total food groups per day. Abbreviations: SCFA, short-chain fatty acids.

## Data Availability

Strauss, Jaclyn; Haskey, Natasha; Ramay, Hena; Ghosh, Tarini; Taylor, Lorian; Yousuf, Munazza; Ohland, Christina; McCoy, Kathy; Ingram, Richard; Ghosh, Subrata; Panaccione, Remo; Raman, Maitreyi (2023), “Weighted gene co-expression network analysis identifies a functional guild and metabolite cluster mediating the relationship between mucosal inflammation and adherence to the Mediterranean diet in Ulcerative Colitis”, Mendeley Data, V1, doi:10.17632/rt3682s6kn.1.

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
