# Peer review of "Weighted Gene Co-Expression Network Analysis Identifies a Functional Guild and Metabolite Cluster Mediating the Relationship between Mucosal Inflammation and Adherence to the Mediterranean Diet in Ulcerative Colitis"

_ijms, 2023, doi:10.3390/ijms24087323_

Round 1

Reviewer 1 Report

This is a good and relevant clinical study with a holistic analysis of the IBD patient cohorts based on FCP as a marker, and then associating their microbiome profiles and a Mediterranean diet.

There are a few questions that require clarification or are of interest:

1. The correlation between the modules versus inflammation and MDS (Fig 7A) seems a little puzzling. The black module seems to be lower associated with inflammation, with no association with MD. The Red module, though points out to beneficial microbes with a lower inflammatory profile, is situated in the center-right quadrant which suggests a tendency to have increased inflammation. The green module seems to be most beneficial, but lacks microbiota correlation. The network analyses should perhaps be "scaled" better to represent your CAG based findings.

2. SCFA (eg, butyric acid) producing microorganisms are associated with stronger barrier function and have been suggested or touted as gut-beneficial flora, and associated with lower inflammation. This seems to be different from blue, brown and black modules associated with microbiome. Perhaps SCFA's could be broken into its own module, from the other metabolites, which are both metabolites of the microbes and the host?

3. Is there a primary associative analysis that can be done with any other biomarker of IBD (Lactoferrin, for eg.) to check if the patients fall into similar cohorts based on that biomarker level? Or even valerate and bile acid that were discovered to be different at week 8, by the authors? This is to check if authors exhausted all options to justify categorization based on FCP only.

4. Although the authors have dissected what specific components of the Mediterranean diet correlates with lower FCP and more favorable microbiota, in the discussion, it would be interesting to add to it what dietary components could be optimized in such a diet, which, based on their overall association between MD and FCP, could make outcomes better?

Reviewer 2 Report

This is an interesting analysis of the impact of a mediteranean diet on inflammation and gut microbiome in a UC population that had been submitted to a dietary intervention for 8 weeks. Despite the original RCT results not showing significant impact on inflammation, there were certain variation in FCal levels that have driven further research  into factors that may predict/drive inflammation. 

Weighted gene co-expression network analysis is a bioinformatics application that identifies modules of highly correlated genes and may suggest the processes that those genes are involved in.

I have a few issues with the design of the study that I will detail below:

- the introduction is not very clear, I couldn't tell clearly what was I going to read about

-the authors then proceed to methods where they detail the methodology of the RCT and nothing on WGCNA; i understand that this is a post hoc analysis of that study, however I still can't tell if I am reading about the RCT or about the WGCNA. Is like they are trying to combine 2 studies into 1 and it is confusing for everyone  

- please give more details on how you performed WGCNA - number of samples, gene clustering, pathway analysis

- with regards to the results, it is stated that 29 patients  with matching gut microbiomes and FCal increase or decrease during the RCT were included in the WCGNA analysis. However, under the paragraph detailing WGCNA it is stated that the analysis had been performed on 26 not the 29 who had variations in FCal levels

- The authors agree that there were no significant differences between medical therapies and there was no therapy change, although they state at the beginning that patients with active disease had been treated accordingly.From what I can see in table 1, the 16 patients with a decrease in FCal were also on treatment in a higher number than those with increase in Fcal. Difficult to assess whether patients have already responded to another therapy - i'd like to see a comment on how flare medication may have had had an influence on results. 

In conclusion:

- I would suggest highlighting more the WCGNA in introduction, as well as in methods, and be more concise on the RCT design

- please address how medication would impact microbiome and therefore study results; this stems from the heterogeneity of the study population, as patients are a mix of active disease and remission

-have the authors considered looking at hub genes and calculating AUC and CIs? - for example genes contained in the green module

-as I stated above, the first paragraph of discussions section is the most clear in the entire paper. Starting from that, please build a more concise introduction and methodology. 

Reviewer 3 Report

Author attempt to enrich and evaluate the functional group and metabolites mediating the correlation between the mucosal inflammation and adherence to the mediterranean diet within the participants diagnosed with ulcerative colitis.

The study approach is novel and the finding is attractive. There is no major flaw found within the manuscript. There are several issue with reference (e.g line 40, 67). The manuscript are ready to be accept as it is once the references is fix.
